# Unraveling the unbinding pathways of SARS-CoV-2 Papain-like proteinase known inhibitors by Supervised Molecular Dynamics simulation

Farzin Sohraby◉, Hassan Aryapour◉*◉

Department of Biology, Faculty of Science, Golestan University, Gorgan, Iran

◉ These authors contributed equally to this work.
* hassan.aryapour@gmail.com

**Data Availability Statement:** Farzin Sohraby, & Hassan Aryapour. (2021). Dataset of the Article "Unraveling the unbinding pathways of SARS-CoV-2 Papain-like proteinase known inhibitors by

## Abstract

The COVID-19 disease has infected and killed countless people all over the world since its emergence at the end of 2019. No specific therapy for COVID-19 is not currently available, and urgent treatment solutions are needed. Recent studies have found several potential molecular targets, and one of the most critical proteins of the SARS-CoV-2 virus work machine is the Papain-like protease (Plpro). Potential inhibitors are available, and their X-ray crystallographic structures in complex with this enzyme have been determined recently. However, their activities against this enzyme are insufficient and need to be characterized and improved to be of clinical values. Therefore, in this work, by utilizing the Supervised Molecular Dynamics (SuMD) simulation method, we achieved multiple unbinding events of Plpro inhibitors, GRL0617, and its derivates, and captured and understood the details of the unbinding pathway. We found that residues of the BL2 loop, such as Tyr268 and Gln269, play major roles in the unbinding pathways, but the most important contributing factor is the natural movements and behavior of the BL2 loop, which can control the entire process. We believe that the details found in this study can be used to refine and optimize potential inhibitors like GRL0617 and design more efficacious inhibitors as a treatment for the SARS-CoV-2 virus.

## Introduction

The emergence of a global pandemic known as the Covid-19 disease caused by the SARS-CoV-2 virus has created close to 45 million infected and 1.2 million deaths until the end of October 2020. This catastrophic event caused unmeasurable damage to both human lives and the global economy [1, 2]. The lack of sufficient treatments has led to an ever-increasing number of deaths, and options are urgently needed [3–6]. Researchers worldwide are working as hard as possible to develop treatment solutions, and since the beginning of the pandemic, several options have been proposed [7–12]. Repurposing existing antivirals such as Remdesivir and Favipiravir have shown antiviral activity against SARS-CoV-2 in clinical trials [8, 13–16]. However, these drugs can only be used for emergencies, and their efficacy is not sufficient.

Supervised Molecular Dynamics simulation". http://doi.org/10.5281/zenodo.4726944.

**Funding:** This investigation was supported by the grant number 99-213-1 from Golestan University, Gorgan, Iran.

**Competing interests:** The authors have declared that no competing interests exist.

Therefore, more selective and more effective treatments are needed to stop and cut the chains of the infected people. Recent studies have shown that one of the most critical proteins of the SARS-CoV-2 virus work machine is the Papain-like protease (Plpro). This cysteine protease can cleave the viral polyproteins (pp1a and pp1ab) into mature functional proteins [17, 18].

Moreover, this enzyme is also responsible for recognizing the Ubiquitin (Ub) and the Ub-like (Ubl) modifier interferon-stimulated gene 15 (ISG15), reversing their mechanism and make the virus escape the immune response of the host cell [19–22]. These make the Plpro an attractive target protein for arresting the virus activity and overcoming this disease. However, firstly, selective and effective inhibitors must be designed.

In the last 15 years since the emergence of SARS-CoV, selective inhibitors of the Plpro of SARS-CoV have been identified with very high inhibitory activities, but they were never considered for further investigations or human use as the virus disappeared. However, The Plpro enzyme of SARS-CoV and the SARS-CoV-2 are very similar in structure and sequence (83% identity) [23]. Therefore, there was always this strong probability that the identified inhibitors of SARS-CoV Plpro can be used against the SARS-CoV-2 Plpro. One of the most potent inhibitors identified originally for the inhibition of SARS-CoV Plpro was GRL0617. In 2008, Ratia et al. introduced naphthalene-based SARS-CoV Plpro inhibitors such as GRL0617 with an $IC_{50}$ value of 0.6 μM [24]. Recently, Osipiuk et al. synthesized derivates of GRL0617 and tested them against the SARS-CoV-2 Plpro, and introduced potent inhibitors [23]. Apart from the GRL0617 ($IC_{50}$ = 2.3 μM), the synthesized compounds had an $IC_{50}$ range of 5.1 to 32.8 μM, which show their great potency.

Furthermore, the 3D crystallographic structure determination of the protein-ligand complexes is a remarkable finding and can immensely accelerate Structure-Based Drug Design (SBDD) investigations for identifying highly selective and highly effective entities to fight the SARS-CoV-2 virus. Although several covalent inhibitors containing aldehyde and Michael acceptor have been identified, they have not sufficient bioavailability and failed in clinical trials [25]. Their undesirable reaction with thiol groups renders them highly toxic despite good activity against cysteine proteases such as Plpro. Therefore, safe, reversible non-covalent small molecule inhibitors are preferred.

Computational Methods in SBDD such as Unbiased Molecular Dynamics (UMD) simulation can accurately investigate the important details of mechanisms such as binding and unbinding mechanisms that are vital for rational drug design [26–32]. Utilizing these fast and accurate methods is of great importance when urgent treatment solutions are needed.

In this work, by utilizing a specialized UMD approach, the supervised MD method, non-covalent inhibitors of SARS-CoV-2 Plpro with available 3D structures were taken into account, and their unbinding pathways were revealed. The main goal of this investigation was extracting the key details governing the efficacy of these inhibitors to gain and report the necessary and valuable information needed for designing more selective and more effective drug candidates of clinical value.

## Method

There are only four unique protein-ligand complexes of Plpro in complex with inhibitor available in Protein Data Bank [33], Plpro in complex with GRL0617 (PDB ID: 7CMD) [34], PLP_Snyder441 (PDB ID: 7JN2) [23], PLP_Snyder495 (PDB ID: 7JIT) [23] and PLP_Snyder530 (PDB ID: 7JIW) [23]. The 3D crystallographic structures were obtained and then cleaned and prepared by UCSF Chimera software [35]. All of the unnecessary molecules, such as water molecules, were deleted from the structures, and the protein-ligand complexes' structures were ready for the next step, the MD simulations. All of the MD simulations were done

by GROMACS 2018 package [36] and OPLS-AA force field [37]. The 3D structures of the co-crystallized inhibitors were parameterized using ACEPYPE [38] with the default setting for assigning the partial charges and atom types. For the construction of the simulation systems, first, the related protein-ligand complex was placed in the center of a triclinic box with a distance of 1 nm from all edges and then solvated with TIP3P water model [39]. Then, sodium and chloride ions were added to produce a neutral physiological salt concentration of 150 mM. Each system was energy minimized, using the steepest descent algorithm, until the Fmax was less than 10 kJ.mol$^{-1}$.nm$^{-1}$. All of the covalent bonds were constrained using the Linear Constraint Solver (LINCS) algorithm [40] to maintain constant bond lengths. The long-range electrostatic interactions were treated using the Particle Mesh Ewald (PME) method [41], and the cut-off radii for Coulomb and Van der Waals short-range interactions were set to 0.9 nm for the interaction of the protein-ligand complex. The modified Berendsen (V-rescale) thermostat [42] and Parrinello–Rahman barostat [43] respectively were applied for 100 and 300 ps to keep the system in the stable environmental conditions (310 K, 1 Bar). Finally, SuMD simulations were carried out under the periodic boundary conditions (PBC), set at XYZ coordinates to ensure that the atoms had stayed inside the simulation box, and the subsequent analyses were then performed using GROMACS utilities, VMD [44] and USCF Chimera, and also the plots were created using Daniel's XL Toolbox (v 7.3.2) add-in [45]. The free energy landscapes were rendered using Matplotlib [46]. In addition, to estimate the binding free energy we used the g_mmpbsa package [47]. The results of the MMPBSA method for each replica and each protein-inhibitor complex is represented as protein-ligand interaction energies. The contribution of each residue during the simulations was calculated by the sum of VdW and electrostatic interaction energies between important residues and the inhibitors during the unbinding pathway. The data was extracted from the trajectory and energy files of simulation using Gromacs's modules such as "gmx rerun" and "gmx energy".

The SuMD simulations were divided into a series of short simulations, called replicas, with a duration time of only 500 ps. The starting point of every series is the crystallographic conformation of the co-crystallized inhibitor in the binding pocket. Then, two points are defined, one was the Center of Mass (COM) of the inhibitor, and the other was set to be the COM of two residues in the deep parts of the binding pocket, Y258 and T259. The distance between these two points was checked at the end of each short 500 ps run. The frame with the highest distance was then selected as the starting point of the next 500 ps simulation. This procedure was repeated until the distance values went above 4 nm, which is a direct sign of the unbinding event.

## Results & discussion

### SuMD simulation of the unbinding pathways

For each of the four inhibitors, three series of SuMD simulations were performed, and all of them were successful, and unbinding events were achieved in the nanosecond time-scale (Table 1). In terms of duration times of the unbinding pathways, the more potent GRL0617 showed better performance than the others and spent more time in the binding pocket, except for the PLP_Snyder530 and the already published activity assays also suggest the same. However, comparing the duration times of the unbinding events may not be a correct way to compare the potency of the inhibitors.

The Plpro has four major domains, the N-terminal ubiquitin-binding domain, the α-helical thumb domain, the β-stranded finger domain, and the palm domain (Fig 1). The Catalytic active site is located in the palm domain where the catalytic triad, Cys111, His272, and Asp286 are placed [34, 48]. The binding site of the GRL0617 and its derivates is also in the palm

**Table 1. The duration times of the unbinding pathways of the inhibitors (ns) with respect to their activity.**

| Inhibitors | Replica series No 1 | Replica series No 2 | Replica series No 3 | experimental activity (μM) [23] |
|---|---|---|---|---|
| GRL0617 | 121 | 72 | 94 | 2.3 |
| PLP_Snyder441 | 166.5 | 75.5 | 31 | NA |
| PLP_Snyder495 | 16 | 24 | 40 | 5.1 |
| PLP_Snyder530 | 152.5 | 77.5 | 57.5 | 6.4 |

domain. However, they do not interact with the catalytic triad and only occupy a fraction of the catalytic active site. There is an essential loop called the BL2 loop in the palm domain, which recognizes the substrate. This loop makes room for substrate and keeps it stable during the cleaving reaction [23, 34, 49–51]. The known inhibitors of SARS-CoV-2 Plpro mainly interact with this loop as this loop can construct a binding site by folding and contracting towards the center of the enzyme. In the crystallographic conformations of the inhibitors, there are stable interactions between the ligands and the residues of the binding site. Important residues such as Tyr 264, Tyr268, and Gln269 on the tip of the BL2 loop can make Pi-stacking and hydrogen bonds, respectively, with this set of inhibitors. By inspecting the results, we found that Gln269 makes these inhibitors more stable in the binding pocket, whereas Tyr268 can control them. GRL0617 and its derivates are naphthalene-based compounds, and their aromatic interaction with Tyr268 is a contributing factor of their unbinding process, which

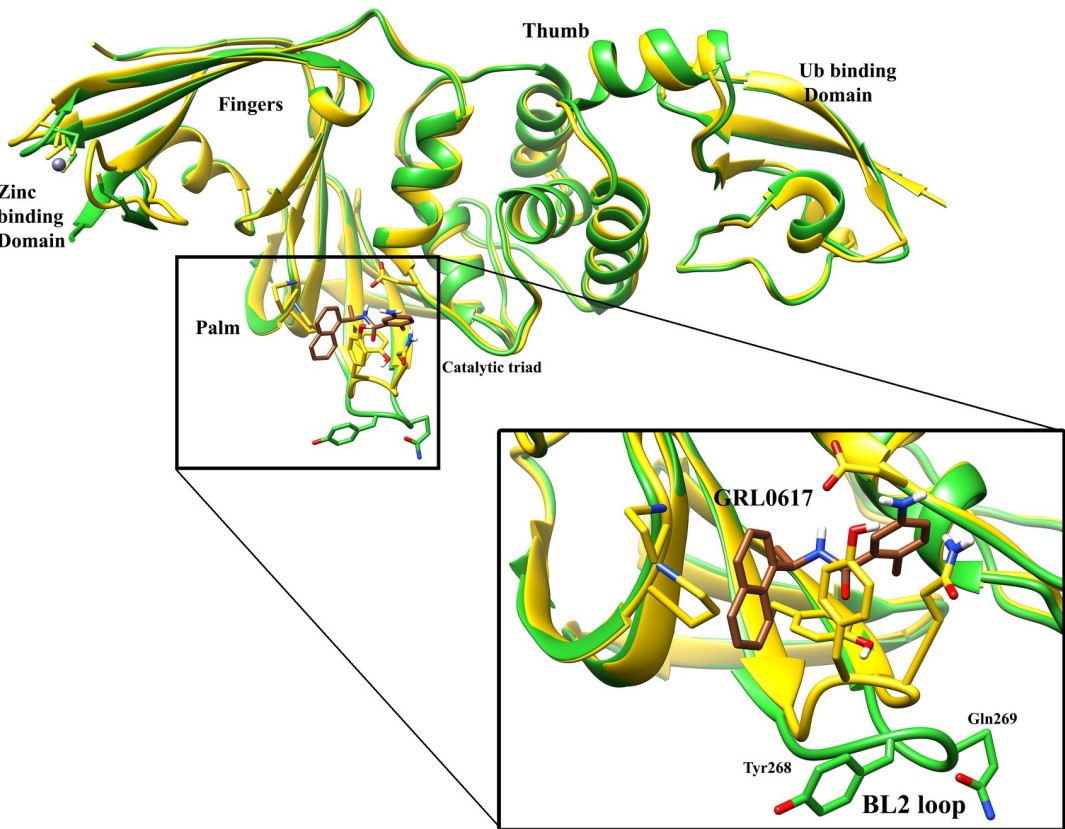

**Fig 1. Structure of SARS-Cov-2 Plpro enzyme.** The superimposed crystallographic structures of the apo form (PDB ID: 6W9C), colored in green, and the complexed form of SARS-CoV-2 Plpro in complex with GRL0617 (PDB ID: 7CMD) [34] colored in yellow. The sequence from Thr266 to Gly272 is considered as the BL2 loop.

will be discussed later. In the next section, the details achieved from the unbinding pathways of each inhibitor are explained.

In the GRL0617-Plpro complex, GRL0617 showed to be one of the most stable inhibitors. In all three series of replicas performed, two stable hydrogen bonds with the Gln269 were observed, one with the nitrogen atom of the backbone and one with the nitrogen atom of the side chain of Gln269 (Fig 2D). The nitrogen atom in the benzamide moiety and the oxygen atom in the center of GRL0617 (N01 and O7) can make hydrogen bonds (Fig 2B, S1 Fig). In all replicas, it was observed that the Tyr268 changes the orientation of its side chain and affects the conformation of the inhibitor. This residue on the tip of the BL2 loop can take the inhibitor out of its crystallographic conformation and lift the naphthalene moiety of the inhibitor. The changes in the orientation of the naphthalene moiety and the Tyr268 residue revealed this fact (Fig 2E and 2D). Then, water molecules could quickly get underneath the inhibitor and disrupt the interaction between the inhibitor and the residues of the binding site, especially the hydrogen bonds, and lifted it (Fig 2F) until it ultimately got out of the binding pocket (Fig 2G). The RMSD values of the inhibitor throughout the simulations (Fig 2A) indicated that it only takes very short amounts of time for the inhibitor to get from the crystallographic conformation to the unbound state, and the intermediate states in-between are not stable and do not last much. The total interaction energies of the protein-ligand complexes (Fig 2C), which are the sum of Lenard-Jones and electrostatic interaction energies, also suggest the same. However, in replica No 2, the interactions between the inhibitor and the protein got weaker, and simultaneously the structure of the BL2 loop got altered at about 40 ns (Fig 2A and 2C). The BL2 loop seems to play a very critical role in the unbinding process. In almost all replicas, the conformation alteration of this loop happened before the unbinding events of the inhibitor. The backbone RMSD values of the residues of the BL2 loop (Fig 2A) illustrate this role. A more effective inhibitor must be able to stop this loop from its natural fluctuations. Additionally, the Free Energy landscape analysis of the unbinding events (Fig 2H–2J) also showed that the intermediate states between the bound state and the unbound states are not stable, and the only stable state is the inhibitor in the crystallographic (native) conformation in which the protein-ligand contact surfaces values are at maximum, and the ligand RMSD values are at their minimum. The movies of the unbinding pathway of the inhibitors can be found in the supplementary materials.

In the PLP_Snyder441-Plpro complex, the duration times of the unbinding events were considerably shorter than the GRL0617, despite the fact that the only difference is the position of the amid group on the benzamide moiety (Fig 3B). Like the GRL0617, two hydrogen bonds with the backbone and the side chain of Gln269 were observed (Fig 3D). In all of the three replicas performed, it was observed that the movements of the BL2 loop affected the inhibitors (Fig 3A). In the native form, the BL2 loop folds towards the center of Plpro, and it has a curved shape. In the SuMD simulations, it was observed that the BL2 gradually took some distance from the enzyme and became flat. It also dragged the inhibitor away from the binding site since the inhibitor can make strong interactions with the residues on the tip of the BL2 loop (Fig 3E and 3F). In this conformation of the BL2 loop, the inhibitor is fully exposed to water molecules and very vulnerable. Then, the interactions got broken in a short amount of time, and the inhibitor got unbound (Fig 3G). This mode of the unbinding pathway only happened in replica No 2 and 3, where the flatting motion of the BL2 loop was observed. The early occurrence of this motion led to the short unbinding event in replica No 2 and 3, about 70 and 30 ns, respectively. The total interaction energy values (Fig 3C), especially in replica No 2, in which the BL2 loop flattened, showed that this motion of the BL2 loop dramatically decreased the interaction energies and made it easier for the inhibitor to unbind. FES analysis of the three replicas (Fig 3H–3J) also illustrated the intermediate state where the inhibitor sticks to

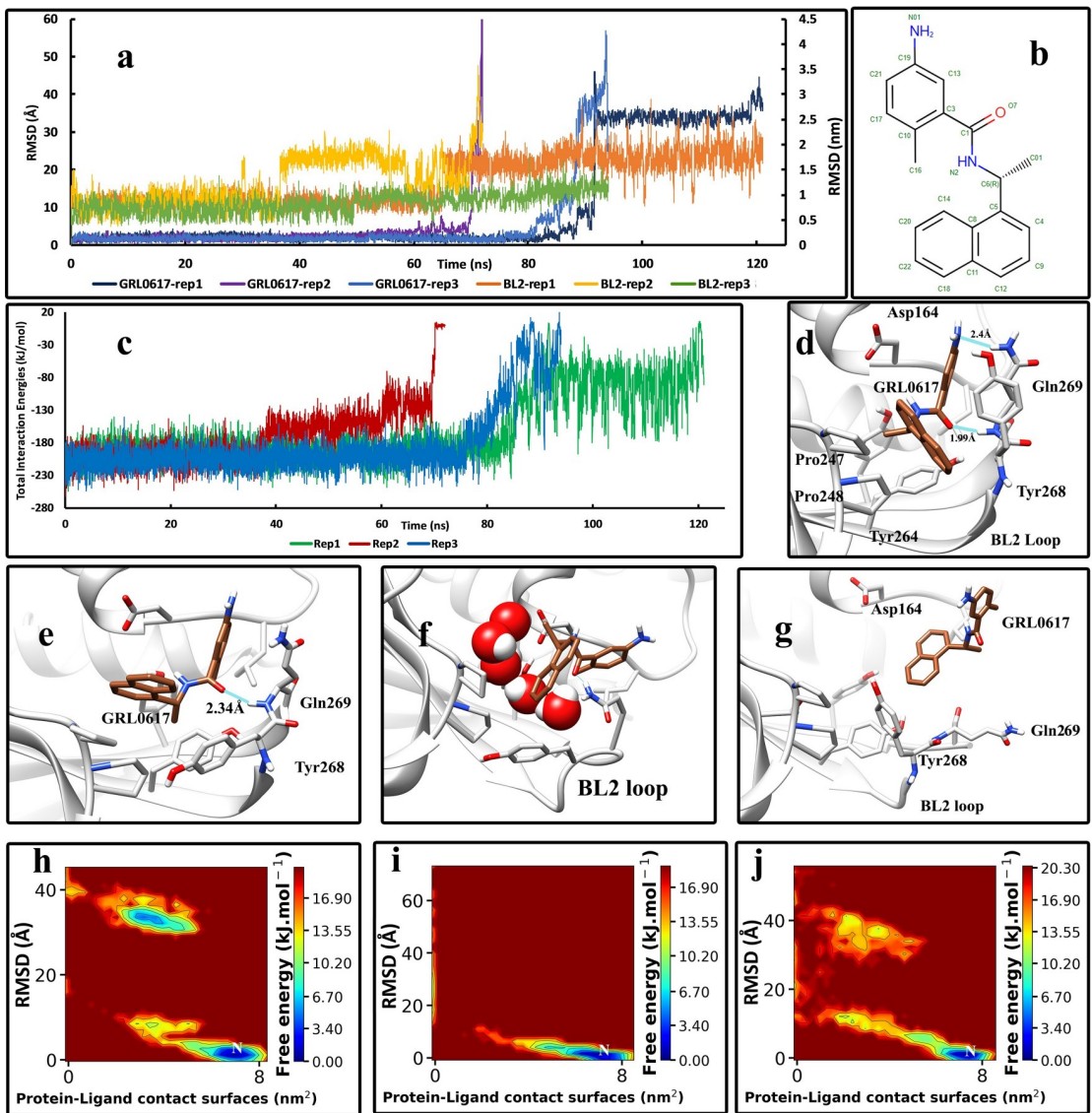

**Fig 2. The details of the unbinding pathway of GRL0617 in three series of replicas. (A)** The RMSD values of the inhibitor in three replicas (Displayed in Å) and the backbone RMSD values of residues of the BL2 loop in three replicas (Displayed in nm) throughout the simulations. **(B)** The 2D structure of GRL0617 was obtained from PDB. **(C)** The Total interaction Energies of the protein-ligand complexes throughout the simulations. **(D)** The native state of the GRL0617 in the crystallographic conformation and the interactions with the residues of the binding pocket (frame at 0 ns). **(E)** An intermediate state of GRL0617 in the unbinding pathway where the Tyr268 residue forced the inhibitor to change its native conformation (2nd replica, frame in 68 ns). **(F)** In another intermediate state, the entire molecule is lifted, and almost all essential bonds and interactions between the ligand and the residues are water-mediated and broken (3rd replica, frame in 85 ns). **(G)** The unbound state of the inhibitor is entirely free and solvated in the simulation box (1st replica, frame in 100 ns). **(H)** The free energy landscape (FEL) representation of the unbinding pathway of the GRL0617, replica No 1. **(I)** The free energy landscape (FEL) representation of the unbinding pathway of the GRL0617, replica No 2. **(J)** The free energy landscape (FEL) representation of the unbinding pathway of the GRL0617, replica No 3. Letter "N" indicates the native crystallographic conformation.

the flattened BL2 loop. This intermediate state is not stable, and the interaction energies are not sufficient to hold the inhibitor.

In the PLP_Snyder495-Plpro complex, the unbinding events happened in very short amounts of time, ranging from 15 to 40 ns (Fig 4A). The addition of two carbamoyl groups to benzamide moiety in the structure of PLP_Snyder495 (Fig 4B) enables it to make extra

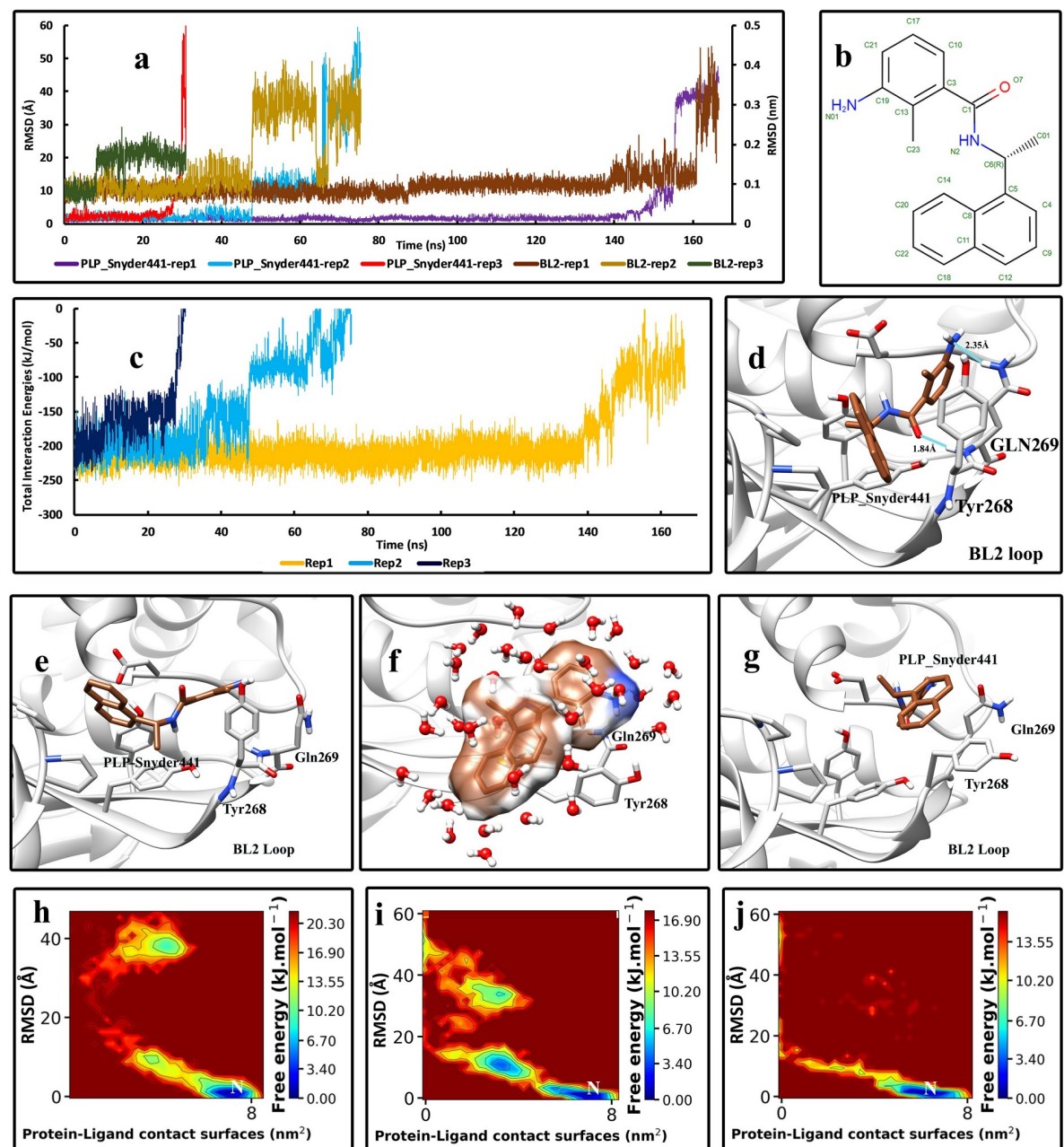

**Fig 3. The details of the unbinding pathway of PLP_Snyder441 from SARS-CoV-2 Plpro in three series of replicas. (A)** The RMSD values of the inhibitor in three replicas (Displayed in Å) and the backbone RMSD values of residues of the BL2 loop in three replicas (Displayed in nm) throughout the simulations. **(B)** The 2D structure of PLP_Snyder441 was obtained from PDB. **(C)** The Total interaction Energies of the protein-ligand complexes throughout the simulations. (D) The bound state of the PLP_Snyder441 in the crystallographic (native) conformation and the interactions with the residues of the binding pocket (frame at 0 ns). **(E)** An intermediate state of PLP_Snyder441 in the unbinding pathway where the BL2 loop gradually starts to take some distance from the enzyme (1st replica, frame in 150 ns). **(F)** Another intermediate state where the BL2 loop is entirely flat and the inhibitor is stuck to the residues on the tip of the loop (Tyr268 and the Gln269) and is almost solvated (2nd replica, frame in 55 ns). **(G)** The unbound state of the inhibitor is entirely free and solvated in the simulation box. **(H)** The free energy landscape (FEL) representation of the unbinding pathway of the PLP_Snyder441, replica No 1. **(I)** The free energy landscape (FEL) representation of the unbinding pathway of the PLP_Snyder441, replica No 2. **(J)** The free energy landscape (FEL) representation of the unbinding pathway of the PLP_Snyder441, replica No 3. Letter "N" indicates the native crystallographic conformation.

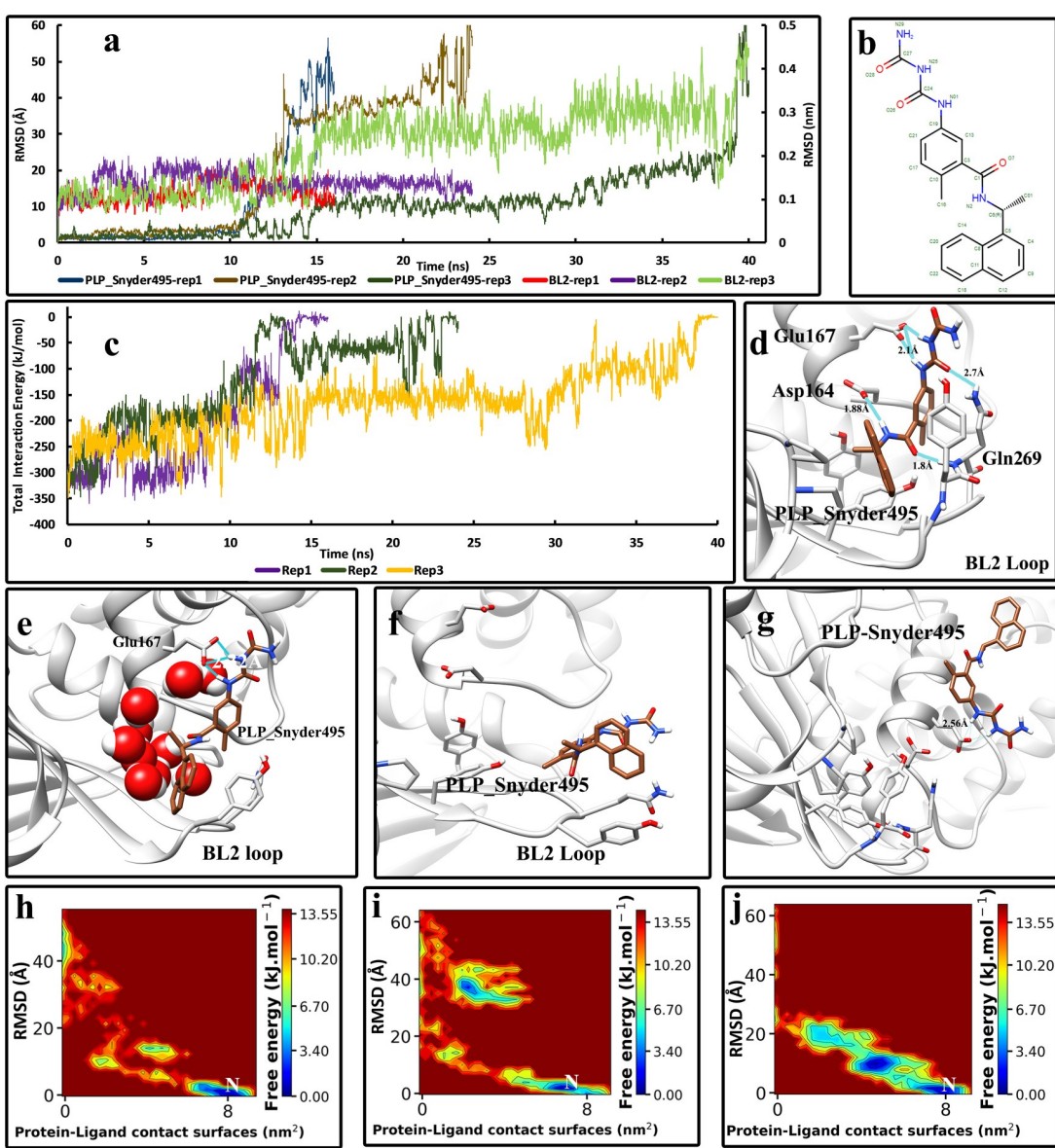

**Fig 4. The details of the unbinding pathway of PLP_Snyder495 from SARS-CoV-2 Plpro in three series of replicas. (A)** The RMSD values of the inhibitor in three replicas (Displayed in Å) and the backbone RMSD values of residues of the BL2 loop in three replicas (Displayed in nm) throughout the simulations. **(B)** The 2D structure of PLP_Snyder495 was obtained from PDB. **(C)** The Total interaction Energies of the protein-ligand complexes throughout the simulations. **(D)** The bound state of the PLP_Snyder495 in the crystallographic (native) conformation and the interactions with the residues of the binding pocket (frame at 0 ns). **(E)** An intermediate state of PLP_Snyder495 in the unbinding pathway where the inhibitor gradually lifted up and out of the binding pocket and made stronger hydrogen bonds with the Glu167 (1st replica, frame in 12 ns). **(F)** Another intermediate state where the BL2 loop is entirely flat and the inhibitor is stuck to the residues on the tip of the loop (Tyr268 and the Gln269) (3rd replica, frame in 25 ns). **(G)** The unbound state of the inhibitor is entirely free and solvated in the simulation box. **(H)** The free energy landscape (FEL) representation of the unbinding pathway of the PLP_Snyder495, replica No 1. **(I)** The free energy landscape (FEL) representation of the unbinding pathway of the PLP_Snyder495, replica No 2. **(J)** The free energy landscape (FEL) representation of the unbinding pathway of the PLP_Snyder495, replica No 3. Letter "N" indicates the native crystallographic conformation.

hydrogen bonds with the residues which are not on the BL2 loop, such as Glu 167 (Fig 4D). Like the previous inhibitors, hydrogen bonds were observed with the residues of the BL2 loop in the crystallographic conformation. Although the hydrogen bonds with the Glu167 are

strong and can hold the inhibitor, the flatting motion of the BL2 loop can still drag the inhibitor away from the binding pocket (Fig 4E and 4F). On the other hand, these extra hydrogen bonds can even force the inhibitor to lift, and in the meanwhile, the interactions can get water-mediated (Fig 4E). These events altogether caused the inhibitor to get out of its native conformation and unbind eventually (Fig 4G). The FES analysis also showed that the inhibitor occupies two intermediate states during the unbinding pathway. It can either be stuck to the tip of the BL2 loop or lift and maintain its hydrogen bonds with the Glu167 (Fig 4H–4J).

In the last complex, PLP_Snyder530-Plpro, in replica No 1 and 2, the conformational changes of the BL2 loop are considerably less than replica No 1. the duration times of the unbinding events was nearly the same as the GRL0617-Plpro complex (Fig 5A). The two inhibitors are very similar in structure and behavior, although the PLP_snyder530 has an acryloylamino group instead of a single amid group on the benzamide moiety (Fig 5B). The interactions present in the native conformation were also the same, but with the difference that in the PLP_Snyder530, the oxygen atom of the acryloylamino group makes a hydrogen bond with the side chain of the Gln269, which is a stronger bond (Fig 5D). The intermediate states observed during the unbinding pathway were almost the same as the previous inhibitors (Fig 5E and 5F), and the FES analysis also illustrates that these intermediate states are not stable and do not last much (Fig 5H–5J). The same behavior and motions of the BL2 loop were seen until the inhibitor was entirely unbound (Fig 5G). The total interaction energy values also suggest that the conformational alterations of the BL2 loop directly affect the binding affinity of the inhibitor (Fig 5C).

In the crystallographic conformation, residues such as Tyr264, Tyr268, Gln269, Arg166, and Leu 162 have the role of stabilizing the inhibitors and have the most interaction energies with the inhibitors (S2 and S3 Figs). Overall, it was found that the unbinding pathways of these inhibitors from the binding pocket are triggered either by interferences of the residues such as Tyr268 or by the changing conformations of the BL2 loop. In these two factors mentioned above, the natural fluctuations of this family of inhibitors can quickly leave the binding site. The most significant contributing factor discovered in the unbinding pathways of all inhibitors was the roles and the behavior of the BL2 loop and the residues on its tip, Tyr268 and Gln269 (S4 Fig). The great movements of this loop are for recognizing P2-P4 of the LXGG motif of substrate [34]. It was also perceived in the results that these movements are not just for trapping the substrate but also for throwing the product out. The crystallographic structures of the complexed or apo forms of the Plpro enzyme are all determined at 100 K. In these low temperatures, this loop will probably show the closed conformation, and even in the apo form (Fig 1), it is not fully open. The only available crystallographic structure of the apo form of Plpro, determined in the room temperature to this date, was the PDB code 6XG3. Even in this structure, the closed form of the BL2 loop is present. However, in the unbinding pathways, it was observed that this loop opens much more than the reported crystallographic structures (Fig 6A). The RMSF values of the residues (Fig 6B) also showed that some areas of the enzyme, such as the Ub binding domain and the BL2 loop, have high fluctuations. The presence of the ligand in the active site may probably lower these fluctuations (Fig 6B, S5 Fig). In the unbinding pathways, the BL2 loop showed that it could drag the inhibitors away from the binding pocket and make them exposed to the water molecules. The residues of the BL2 loop, such as Tyr268 and Gln269, can also make strong interactions such as hydrogen bonds and Pi-stacking for keeping the inhibitors glued to the tip of the BL2 loop (Figs 3F, 5F and 6F). This path line where the inhibitor stayed connected to the tip of the BL2 loop and the BL2 loop got flattened, was one of the unbinding pathways discovered in this study. We believe that the flexibility of the BL2 loop is a natural behavior, and it certainly works at a specific rate to facilitate the activity of the enzyme, recognizing, trapping, and throwing out the resulting product. Disabling this movement might also be a route to inhibit the activity of the enzyme.

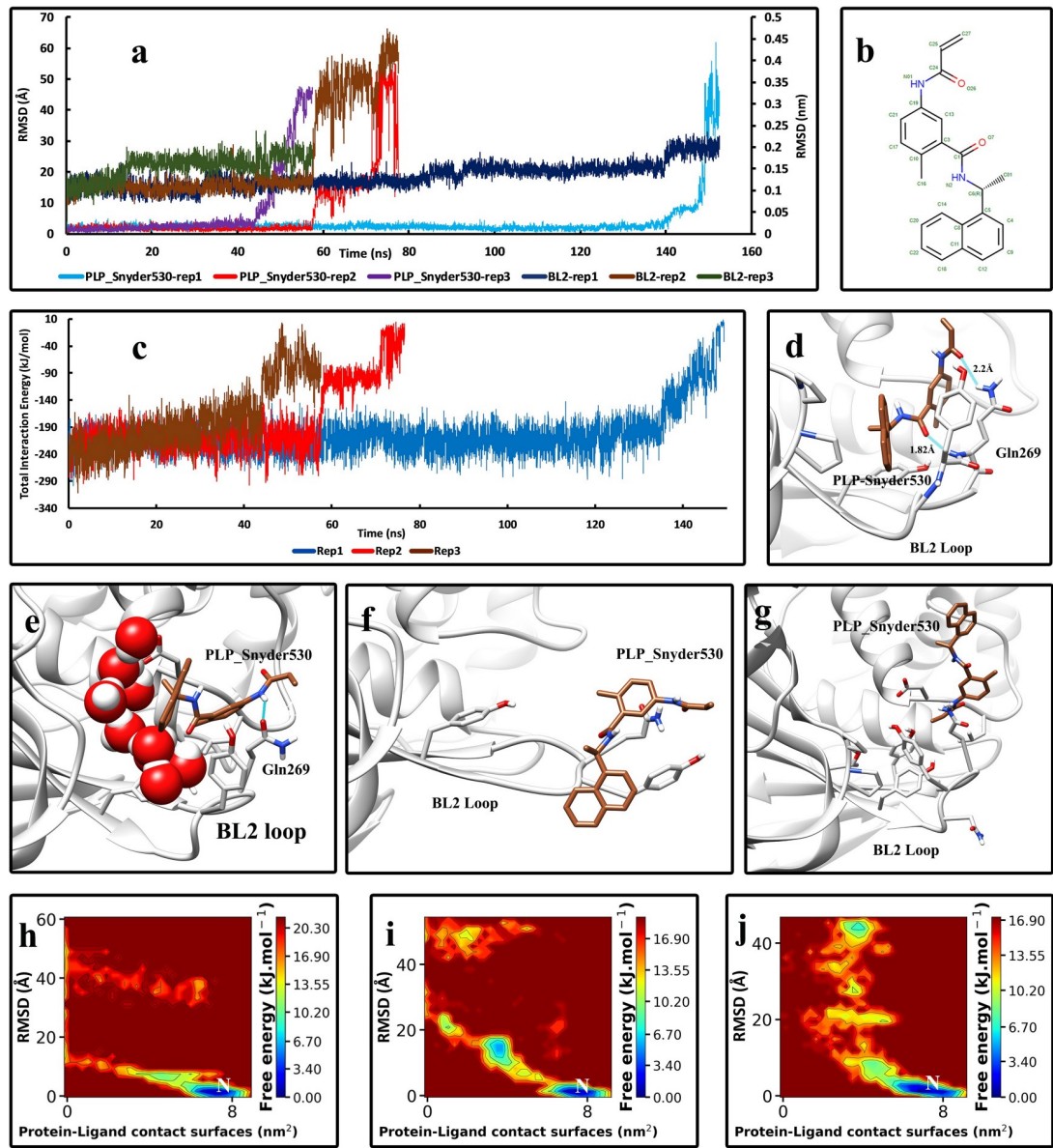

**Fig 5. The details of the unbinding pathway of PLP_Snyder530 from SARS-CoV-2 Plpro in three series of replicas. (A)** The RMSD values of the inhibitor in three replicas (Displayed in Å) and the backbone RMSD values of residues of the BL2 loop in three replicas (Displayed in nm) throughout the simulations. **(B)** The 2D structure of PLP_Snyder530 was obtained from PDB. **(C)** The Total interaction Energies of the protein-ligand complexes throughout the simulations. **(D)** The bound state of the PLP_Snyder530 in the crystallographic (native) conformation and the interactions with the residues of the binding pocket (frame at 0 ns). **(E)** An intermediate state of PLP_Snyder530 in the unbinding pathway in which the naphthalene moiety had turned and is out of the crystallographic (native) binding conformation (3rd replica, frame in 50 ns). **(F)** Another intermediate state where the BL2 loop is completely flat and the inhibitor is stuck to the residues on the tip of the loop (Tyr268 and the Gln269) (2[nd] replica, frame in 60 ns). **(G)** The unbound state of the inhibitor that is completely free and solvated in the simulation box. **(H)** The free energy landscape (FEL) representation of the unbinding pathway of the PLP_Snyder530, replica No 1. **(I)** The free energy landscape (FEL) representation of the unbinding pathway of the PLP_Snyder530 replica No 2. **(J)** The free energy landscape (FEL) representation of the unbinding pathway of the PLP_Snyder530, replica No 3. Letter "N" indicates the native crystallographic conformation.

The main goal of this study was to find the details of the unbinding pathway of Plpro inhibitors and understand the governing factors to find ways to improve these compounds. By analyzing the results and inspecting the protein-inhibitor complexes and the unbinding pathways,

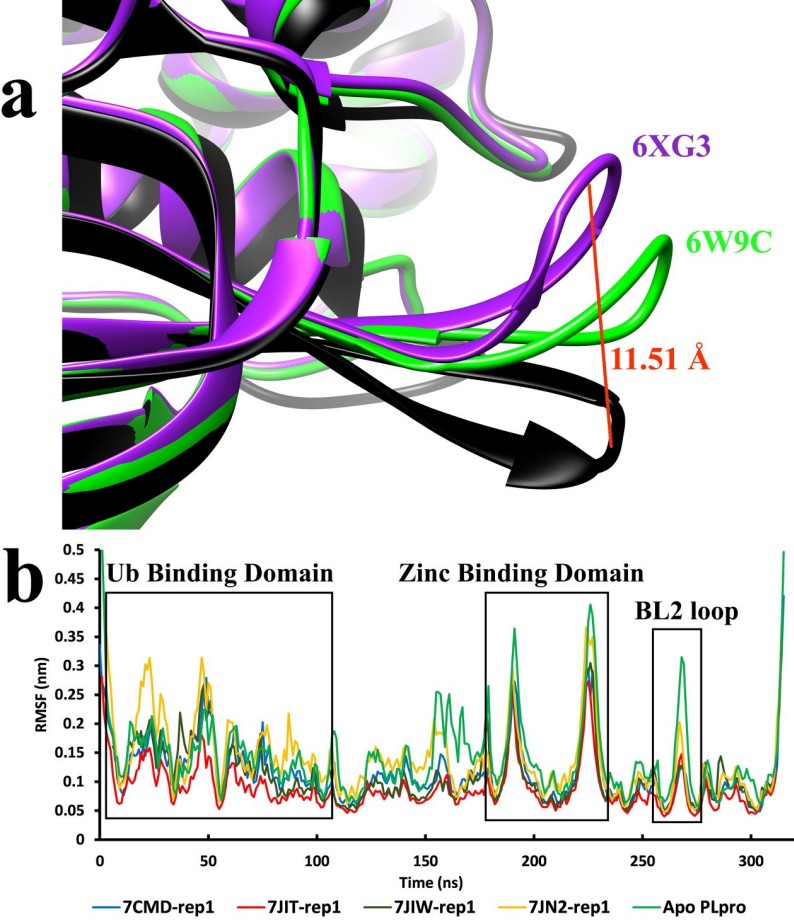

**Fig 6. The flexibility of the BL2 loop.** (A) the crystallographic structures of the Plpro are superimposed to compare the flexibility of the BL2 loop and the degree of its opening mechanism. The fully open structure of the BL2 (Black), which was achieved from the unbinding pathways and in the presence of the inhibitor, showed that the distance between the Cα atoms of the Tyr268 on the tip of the BL2 loop in each structure reaches 11.5 Å. (B) the RMSF values of the residues of the Plpro enzyme during the unbinding pathways of the inhibitors and in the apo form of the enzyme, with respect to the important regions of the enzyme.

we found that although many bonds and interactions are formed between the inhibitors and the residues of the binding pocket, which make them very stable, the natural fluctuation of the BL2 loop can easily displace the inhibitors and make them unbind. One of the best solutions to improve these naphthalene-based inhibitors is adding substitution groups or moieties to increase interaction with regions other than the BL2 loop. If the inhibitors make more interactions with the BL2 loop, it would be easier for this loop to displace them. Therefore, less interaction with the BL2 loop and more bonds and interactions with the inner parts of the binding pocket is needed. Among the synthesized inhibitors by Osipiuk et al. [23], PLP_Snyder495 had extra groups on the benzamide moiety that could make strong hydrogen bonds with the Glu167. Although this residue is not on the BL2 loop and located on the upper side of the binding pocket, it can make strong hydrogen bonds with the PLP-Snyder495 (Fig 4D). However, this residue is fully exposed to water molecules. The hydrogen bonds formed between the inhibitor and this residue can easily get water-mediated and broken; thus, this compound had the worst performance in our simulations and left the binding pocket in very short amounts of time (Table 1). Making strong interactions with the correct residues is the key. All of these

inhibitors can strongly bind to the BL2 loop. We believe that minimizing interactions with the BL2 loop and maximizing interactions with the inner parts of the binding pocket where there is more limited access to water molecules is the best way to improve these inhibitors. GRL0617 has proven to inhibit the cytopathogenic effect of the virus [49]. Optimizing and improving this potent inhibitor is the route to an efficacious treatment for Covid-19. The Details understood from this study can be used for this purpose.

## Conclusion

Plpro has proved to be a tremendous molecular target for fighting SARS-CoV-2. Inhibition of this enzyme by GRL0617 and its derivatives has inhibited the virus's cytopathogenic effect and making them highly potential. However, their efficacy is not sufficient enough, and more effective inhibitors are urgently needed. In this work, the SuMD method enabled us to achieve multiple unbinding events of known Plpro inhibitors and understand vital details that can help improve the efficacy of these inhibitors. We found that the BL2 loop has natural motions that are probably very important in the enzyme activity. Improving the available inhibitors by considering the motions of this loop and the residues involved will produce more effective and more efficacious inhibitors.

## Supporting information

**S1 Fig. The distance between the atoms contributing in important hydrogen bonds in the protein-ligand complexes. (A)** The important hydrogen bonds in the GRL0617-Plpro complex. **(B)** The important hydrogen bonds in PLP-Snyder441-Plpro complex. **(C)** The important hydrogen bonds in PLP_Snsyder495-Plpro complex. **(D)** The important hydrogen bonds in PLP_Snsyder530-Plpro complex.
(TIF)

**S2 Fig. The interaction energy contribution of the residues in contact with the inhibitor in the GRL0617-Plpro complex. A)** replica No 1. **(B)** replica No 2. **(C)** replica No 3.
(TIF)

**S3 Fig. The interaction energy contribution of the residues in contact with the inhibitor in the PLP_Snyder495-Plpro complex. (A)** replica No 1. **(B)** replica No 2. **(C)** replica No 3.
(TIF)

**S4 Fig. The VdW and electrostatic interaction energies of each replica in the unbinding pathways. (A)** GRL0617-Plpro complex. **(B)** PLP-Snyder441-Plpro complex. **(C)** PLP_Snsyder495-Plpro complex. **(D)** PLP_Snsyder530-Plpro complex.
(TIF)

**S5 Fig. The backbone RMSD values of the Plpro enzyme in apo form.** The duration of the simulation was 100 ns.
(TIF)

**S1 File. 7cmd-rep1.mp4, the unbinding pathway of GRL0617 from the Plpro enzyme.**
(MP4)

**S2 File. 7JIT-rep1.mp4, the unbinding pathway of PLP_Snyder495 from the Plpro enzyme.**
(MP4)

**S3 File. 7JN2-rep1.mp4, the unbinding pathway of PLP_Snyder441 from the Plpro enzyme.**
(MP4)

**S4 File. 7JIW-rep1.mp4, the unbinding pathway of PLP_Snyder530 from the Plpro enzyme.**
(MP4)

## Author Contributions

**Conceptualization:** Hassan Aryapour.

**Data curation:** Hassan Aryapour.

**Formal analysis:** Hassan Aryapour.

**Investigation:** Farzin Sohraby, Hassan Aryapour.

**Methodology:** Hassan Aryapour.

**Software:** Farzin Sohraby.

**Supervision:** Hassan Aryapour.

**Validation:** Farzin Sohraby, Hassan Aryapour.

**Visualization:** Farzin Sohraby.

**Writing – original draft:** Farzin Sohraby.

**Writing – review & editing:** Farzin Sohraby, Hassan Aryapour.

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
