## [Decision Letter · Decision Letter 0]

14 Apr 2021

PONE-D-21-08289

Unraveling the unbinding pathways of SARS-CoV-2 Papain-like proteinase known inhibitors by Supervised Molecular dynamics simulation

PLOS ONE

Dear Dr. Aryapour,

Thank you for submitting your manuscript to PLOS ONE. After careful consideration, we feel that it has merit but does not fully meet PLOS ONE’s publication criteria as it currently stands. Therefore, we invite you to submit a revised version of the manuscript that addresses the points raised during the review process.

Please address each of the points raised by the reviewers and note in the cover letter the corresponding revisions of the manuscript. Include in the cover letter detailed responses to the comments of the reviewers.

A rebuttal letter that responds to each point raised by the reviewers. You should upload this letter as a separate file labeled 'Response to Reviewers'.A marked-up copy of your manuscript that highlights changes made to the original version. You should upload this as a separate file labeled 'Revised Manuscript with Track Changes'.An unmarked version of your revised paper without tracked changes. You should upload this as a separate file labeled 'Manuscript'.

We look forward to receiving your revised manuscript.

Kind regards,

Emilio Gallicchio, Ph.D.

Academic Editor

PLOS ONE

Journal Requirements:

5. Please include captions for your Supporting Information files at the end of your manuscript, and update any in-text citations to match accordingly. Please see our Supporting Information guidelines for more information: http://journals.plos.org/plosone/s/supporting-information

Reviewers' comments:

Reviewer's Responses to Questions

**Comments to the Author**

1. Is the manuscript technically sound, and do the data support the conclusions?

Reviewer #1: Partly

Reviewer #2: Yes

2. Has the statistical analysis been performed appropriately and rigorously? 

Reviewer #1: No

Reviewer #2: Yes

3. Have the authors made all data underlying the findings in their manuscript fully available?

Reviewer #1: Yes

Reviewer #2: Yes

4. Is the manuscript presented in an intelligible fashion and written in standard English?

Reviewer #1: Yes

Reviewer #2: Yes

5. Review Comments to the Author

Reviewer #1: Comments for the Authors

The outbreak and spread of COVID-19 diseases caused by the severe acute respiratory syndrome coronavirus 2 (SARS-CoV-2) infection now is well-known as a global concern to the public health worldwide. Most infected people develop asymptomatic symptoms or mild to moderate illness and recover without hospitalization. However, for these of severe cases will have worsening dyspnea accompanied by hypoxemia and further turn to progressive respiratory failure, which can be lethal. Vaccine can prevent the spread of COVID-19 for the uninfected individuals. However, for these SARS-CoV-2 infected patients with acute diseases, pharmaceutic intervention with significant therapeutic efficiency would be an essential antiviral treatment strategy. The papain-like protease (PLpro) of SARS-CoV-2 is a highly versatile enzyme that processes viral polyproteins to generate a functional replicase complex and enable viral spread, also essential for cleaving proteinaceous post-translational modifications on host proteins as an evasion mechanism against host antiviral immune responses. Inhibition of SARS-CoV-2 PLpro with inhibitor could be a potential therapeutic strategy through impeding viral replication in infected cells. In this study, Farzin et al. performed Supervised Molecular Dynamics (SuMD) simulations to decipher the unbinding pathway of SARS-CoV-2 PLpro and its four inhibitors including GRL0617 and its derivates. The simulation stability of SARS-CoV-2 PLpro and GRL0617 outperforms those of the other three, in agreement with the experimental IC50 data. The 2D free energy changes caused by the binding of inhibitors were calculated. Their respective intermediate binding poses, protein-ligand interactions such as hydrogen bond changes over time, RMSF of PLpro residues at holo state were analyzed. In summary, the work is perspectively important and properly descripted. It provides useful conclusions about how GRL0617 and its derivates disassociated with the PLpro binding, in particular, the major role of Tyr268 and Gln269 residues of the BL2 loop in the unbinding pathways.

There are several points described below,

1.For each inhibitor, 3 series of SuMD simulations were performed. However, the 2D free energy profiles are likely not converged, as new minima emerged in each of three maps, for instance, Fig. 5h-j. More series of SuMD simulations should be needed.

2. Although the authors mentioned binding free energy calculation by MM/PBSA method, however, I did not find the calculation results. Also, the authors also estimated the contribution of each amino acids, as shown in Supplementary Figs. 2-3, however, the authors did not describe in the manuscript about how these contributions were estimated, by free energy decomposition method?

3.I suggest to highlight the catalytic triad and the BL2 loop with different color in the upper panel of Fig. 1, while yellow colored residues interacting with GRL0617 ligand should be labeled too in the lower panel of Fig. 1. In addition, BL2 loop should be defined, from which residue to which residue?

4.(1)What is the reference structure used for RMSD calculation in Figs. 2-4a?

(2)I suggest to split the total interaction so as to also include the ele and vdw interactions in Figs. 2-4c to show their changes.

(3)The distance of hydrogen bonds in Figs. 2-4 should be labeled.

(4)There are many intermediate states during a MD simulation, which frames were extracted as Figs. 2-4e,f? Please clarify. It is important to find these key intermediate states correctly for each inhibitor, so that their comparisons were fair.

(5)Did the author extract out the structures from the minima of three free energy landscapes in Figs. 2-4h-j? These metastable structures should be compared and discussed.

(6)For each inhibitor, did three series SuMD show the same trajectory path line? I suggest the authors provide three trajectory path lines with three colors, to see if the inhibitor exit the binding pocket with different pathways.

(7)The crystallographic (native) conformation should be marked in Fig.s 2-4h-j.

5.For PLP_Snyder441-Plpro complex, the authors mentioned that “In all of the three replicas performed, it was observed that the movements of the BL2 loop had a direct effect on the inhibitors (Fig. 3a).” However, as shown in Fig. 3a, the replica 3 seems an opposite case. Because although BL2 dramatically shifted at ~8 ns, PLP_Snyder441 looked stable over the whole MD time. The reason should be tracked by analyzing the extracted structures.

6.The authors should run a direct MD simulation for SARS-CoV-2 PLpro at apo state, and calculate RMSF for each residue to show the nature flexibility particularly for BL2 loop to support the point that “The presence of the ligand in the active site may probable lower these fluctuations”.

7.In Fig. 6, the binding ligands of crytal structure (green) and MD structure (black) should be presented using the same color with their corresponding proteins.

Minor issues:

1. “The RMSD values of the inhibitor throughout the simulations (Fig. 2a) indicated that it only takes very short amounts of time for the inhibitor to get from the crystallographic conformation to the unbound state”, the “amounts” here is subscript, please correct.

2. Please provide legends for movies as well. The movies should be cited in the main text properly.

3. “On the other hand, these extra hydrogen bonds can even force the inhibitor to lift up and in the meanwhile the interactions can get water-mediated (Fig. 4c).” “Fig. 4c” should be “Fig. 4e”.

Reviewer #2: The article is straightforward and for most of the parts clear. The impact that it would provide is significant since it provides insight into potential drug discovery against COVID-19. I therefore believe that it is valuable, and it should be published. However, it needs some revisions:

1. In Table 1 the authors claim “In terms of duration, the more potent GRL0617 showed better performance than the others, except for the PLP_Snyder530, and the already published activity assays also suggest the same. However, comparing the duration times of the unbinding events is not a correct way to compare the potency of the inhibitors.”

If “comparing the duration times of the unbinding events is not a correct way to compare the potency of the inhibitors”, what is the point of even including Table 1?

2. In figure 1, none of the ligand-receptor interactions (H-bonds and pi-stacking) are depicted.

3. In all the figures, I notice inconsistencies: the BL2 is not always labelled, some key residues are not always labelled, pi-stacking interactions are never depicted.

4. In the last portion of the discussion I notice another contradiction. The authors claim: “Among the synthesized inhibitors by Osipiuk et al.[23], PLP_Snyder495 had extra groups on the benzamide moiety that could make strong hydrogen bonds with the Glu167 which is not on the BL2 loop. However, this residue is fully exposed to water molecules and the hydrogen bonds can easily get water-mediated and broken and as a result, this compound had the worst performance in our simulations and left the binding pocket in very short amounts of time (Table 1). We believe that minimizing interactions with the BL2 loop and maximizing interactions with the inner parts of the binding pocket where there is more limited access to water molecules is the best way to improve these inhibitors.”

Glu 167 seems to be in the inner part of the pocket and yet the molecule does not bind strongly enough. This seems to contradict the last sentence. Can you please explain?

6. PLOS authors have the option to publish the peer review history of their article (what does this mean?). If published, this will include your full peer review and any attached files.

Reviewer #1: No

Reviewer #2: **Yes: **Pierpaolo Cordone

---

## [Author Response · Author response to Decision Letter 0]

29 Apr 2021

Dear Reviewers;

Thank you for reviewing our manuscript. We appreciate your constructive comments for improving the article. Your comments have been fully applied to the text and the figures of the manuscript. It is a great honor to be a part of your prestigious journal.

Sincerely,

Hassan Aryapour

Comments to the Author

Answers were added

Reviewer #1: The outbreak and spread of COVID-19 diseases caused by the severe acute respiratory syndrome coronavirus 2 (SARS-CoV-2) infection now is well-known as a global concern to the public health worldwide. Most infected people develop asymptomatic symptoms or mild to moderate illness and recover without hospitalization. However, for these of severe cases will have worsening dyspnea accompanied by hypoxemia and further turn to progressive respiratory failure, which can be lethal. Vaccine can prevent the spread of COVID-19 for the uninfected individuals. However, for these SARS-CoV-2 infected patients with acute diseases, pharmaceutic intervention with significant therapeutic efficiency would be an essential antiviral treatment strategy. The papain-like protease (PLpro) of SARS-CoV-2 is a highly versatile enzyme that processes viral polyproteins to generate a functional replicase complex and enable viral spread, also essential for cleaving proteinaceous post-translational modifications on host proteins as an evasion mechanism against host antiviral immune responses. Inhibition of SARS-CoV-2 PLpro with inhibitor could be a potential therapeutic strategy through impeding viral replication in infected cells. In this study, Farzin et al. performed Supervised Molecular Dynamics (SuMD) simulations to decipher the unbinding pathway of SARS-CoV-2 PLpro and its four inhibitors including GRL0617 and its derivates. The simulation stability of SARS-CoV-2 PLpro and GRL0617 outperforms those of the other three, in agreement with the experimental IC50 data. The 2D free energy changes caused by the binding of inhibitors were calculated. Their respective intermediate binding poses, protein-ligand interactions such as hydrogen bond changes over time, RMSF of PLpro residues at holo state were analyzed. In summary, the work is perspectively important and properly descripted. It provides useful conclusions about how GRL0617 and its derivates disassociated with the PLpro binding, in particular, the major role of Tyr268 and Gln269 residues of the BL2 loop in the unbinding pathways.

There are several points described below,

Question #1: For each inhibitor, 3 series of SuMD simulations were performed. However, the 2D free energy profiles are likely not converged, as new minima emerged in each of three maps, for instance, Fig. 5h-j. More series of SuMD simulations should be needed.

Answer #1: The FEL profiles may not converge, and it is because each of the inhibitors may have many routes for their unbinding pathways. However, the first steps of unbinding pathways in the three replicas of each inhibitor were almost the same. In each replica, the main player is the fluctuation of the BL2 loop and the two most important residues, Tyr268 and Gln269.

Question #2: Although the authors mentioned binding free energy calculation by MM/PBSA method, however, I did not find the calculation results. Also, the authors also estimated the contribution of each amino acids, as shown in Supplementary Figs. 2-3, however, the authors did not describe in the manuscript about how these contributions were estimated, by free energy decomposition method?

Answer #2: The results of the MMPBSA method for each replica and each protein-inhibitor complex are presented in Fig2.c, Fig3.c, Fig4.c, and Fig5.c, and it is represented as protein-ligand interaction energies. The contribution of each residue during the simulations was calculated by the sum of VdW and electrostatic interaction energies between important residues and the inhibitors during the unbinding pathway. The data was extracted from the trajectory and energy files of simulation using Gromacs’s modules such as “gmx rerun” and “gmx energy”. The related information was added to the “Materials and Methods” section.

Question #3: I suggest to highlight the catalytic triad and the BL2 loop with different color in the upper panel of Fig. 1, while yellow colored residues interacting with GRL0617 ligand should be labeled too in the lower panel of Fig. 1. In addition, BL2 loop should be defined, from which residue to which residue?

Answer #3: The figure 1 does not only illustrate the position of the BL2 loop and the catalytic triad on the structure of the Plpro but showing the difference between the apo form and the complex form of this enzyme in the presence of the inhibitor, GRL0617 in the binding site. However, for more clarification, the range of the residues of the BL2 loop was added to the caption of the figure.

Questions #4.

What is the reference structure used for RMSD calculation in Figs. 2-4a?

(1) The reference for the calculation of the RMSD values is the crystallographic structure of the complexes.

(2) I suggest to split the total interaction so as to also include the ele and vdw interactions in Figs. 2-4c to show their changes.

(2) The divided interaction energies were provided as a supplementary figure, Sup Fig 4. 

(3) The distance of hydrogen bonds in Figs. 2-4 should be labeled.

(3) Corrected

(4) There are many intermediate states during a MD simulation, which frames were extracted as Figs. 2-4e,f? Please clarify. It is important to find these key intermediate states correctly for each inhibitor, so that their comparisons were fair.

(4) Corrected

(5) Did the author extract out the structures from the minima of three free energy landscapes in Figs. 2-4h-j? These metastable structures should be compared and discussed.

(5) Yes. The frames shown in the figures were extracted according to the state of the inhibitor during the unbinding pathway. 

(6) For each inhibitor, did three series SuMD show the same trajectory path line? I suggest the authors provide three trajectory path lines with three colors, to see if the inhibitor exit the binding pocket with different pathways.

(6) Due to the limitation of the SuMD simulation, the path line of the unbinding pathway of each inhibitor was almost the same.

(7) The crystallographic (native) conformation should be marked in Fig.s 2-4h-j.

(7) Added

Question #5: For PLP_Snyder441-Plpro complex, the authors mentioned that “In all of the three replicas performed, it was observed that the movements of the BL2 loop had a direct effect on the inhibitors (Fig. 3a).” However, as shown in Fig. 3a, the replica 3 seems an opposite case. Because although BL2 dramatically shifted at ~8 ns, PLP_Snyder441 looked stable over the whole MD time. The reason should be tracked by analyzing the extracted structures.

Answer #5: in the case of PLP-Snyder441, the structure of the BL2 loop changed significantly, as you mentioned. This structural change led to the water-mediation of the interactions of the inhibitor and the residues and caused it to unbind moments later. It is better to say that in this case, rep3, the movements of the BL2 loop had rather indirect effects on the unbinding pathway of this inhibitor. However, its crucial influence on the unbinding pathway is undeniable.

Question #6: The authors should run a direct MD simulation for SARS-CoV-2 PLpro at apo state, and calculate RMSF for each residue to show the nature flexibility particularly for BL2 loop to support the point that “The presence of the ligand in the active site may probable lower these fluctuations”.

Answer #6: The Apo state of Plpro has been simulated for 100 ns, and RMSF values have been added to Fig 6b.

Question #7: In Fig. 6, the binding ligands of crytal structure (green) and MD structure (black) should be presented using the same color with their corresponding proteins.

Answer #7: the purpose of Fig 6a was to show the severity of the BL2 loop fluctuations. The structures were chosen for this analysis, 6W9C and 6XG3, are the apo forms of the protein, and the black structure is a simulated model of Plpro when the ligand was about to leave the binding pocket and the end of the unbinding pathway. Similar structure can be found in Fig 5f. 

Minor issues:

(1). “The RMSD values of the inhibitor throughout the simulations (Fig. 2a) indicated that it only takes very short amounts of time for the inhibitor to get from the crystallographic conformation to the unbound state”, the “amounts” here is subscript, please correct.

(1) corrected

(2). Please provide legends for movies as well. The movies should be cited in the main text properly.

(2) Added

(3). “On the other hand, these extra hydrogen bonds can even force the inhibitor to lift up and in the meanwhile the interactions can get water-mediated (Fig. 4c).” “Fig. 4c” should be “Fig. 4e”.

(3) Corrected

Reviewer #2: The article is straightforward and for most of the parts clear. The impact that it would provide is significant since it provides insight into potential drug discovery against COVID-19. I therefore believe that it is valuable, and it should be published. However, it needs some revisions:

Question #1: In Table 1 the authors claim “In terms of duration, the more potent GRL0617 showed better performance than the others, except for the PLP_Snyder530, and the already published activity assays also suggest the same. However, comparing the duration times of the unbinding events is not a correct way to compare the potency of the inhibitors. ”If “comparing the duration times of the unbinding events is not a correct way to compare the potency of the inhibitors”, what is the point of even including Table 1?

Answer #1: The point of including table 1 was to show the potency of the inhibitors in wet-lab experiments and their activity against Plpro. However, some correlations can be found between the duration times and the activity of the inhibitors. The sentence in the text of the manuscript has been corrected accordingly.

Question #2: In figure 1, none of the ligand-receptor interactions (H-bonds and pi-stacking) are depicted.

Answer #2: No, none of the interactions are shown. Because the purpose of this figure was only to show the position of the BL2 loop and the binding site of the inhibitors, and the fluctuations and the movements of the BL2 loop in apo form and in the presence of the inhibitor. The protein-inhibitor interactions in the crystallographic form are all shown in the “d” section of each figure.

Question #3: In all the figures, I notice inconsistencies: the BL2 is not always labelled, some key residues are not always labelled, pi-stacking interactions are never depicted.

Answer #3: Only essential labels and details were added to each figure to avoid putting packed labels and make the figures very crowded. All of the essential information needed to understand the data is present in the figures. However, more details and labels have been added in the revision.

Question #4: In the last portion of the discussion, I notice another contradiction. The authors claim: “Among the synthesized inhibitors by Osipiuk et al.[23], PLP_Snyder495 had extra groups on the benzamide moiety that could make strong hydrogen bonds with the Glu167 which is not on the BL2 loop. However, this residue is fully exposed to water molecules and the hydrogen bonds can easily get water-mediated and broken and as a result, this compound had the worst performance in our simulations and left the binding pocket in very short amounts of time (Table 1). We believe that minimizing interactions with the BL2 loop and maximizing interactions with the inner parts of the binding pocket where there is more limited access to water molecules is the best way to improve these inhibitors.”

Glu 167 seems to be in the inner part of the pocket and yet the molecule does not bind strongly enough. This seems to contradict the last sentence. Can you please explain?

Answer #4: As shown in Fig 4d, Glu167 is exposed to water molecules and located on the upper side of the binding pocket, not the inner parts. The inner parts are less exposed. The text of this bit has been changed to make it more understandable and clear.

---

## [Editor Report · Decision Letter 1]

6 May 2021

Unraveling the unbinding pathways of SARS-CoV-2 Papain-like proteinase known inhibitors by Supervised Molecular Dynamics simulation

PONE-D-21-08289R1

Dear Dr. Aryapour,

We’re pleased to inform you that your manuscript has been judged scientifically suitable for publication and will be formally accepted for publication once it meets all outstanding technical requirements.

Kind regards,

Emilio Gallicchio, Ph.D.

Academic Editor

PLOS ONE
---

## [Editor Report · Acceptance letter]

10 May 2021

PONE-D-21-08289R1 

Unraveling the unbinding pathways of SARS-CoV-2 Papain-like proteinase known inhibitors by Supervised Molecular Dynamics simulation 

Dear Dr. Aryapour:

I'm pleased to inform you that your manuscript has been deemed suitable for publication in PLOS ONE. Congratulations! Your manuscript is now with our production department. 

Kind regards, 

on behalf of

Dr Emilio Gallicchio 

Academic Editor

PLOS ONE